# Changes in Clinical Characteristics and Outcomes of Patients Admitted to Inpatient Cardiac Rehabilitation

**DOI:** 10.3390/ijerph18168871

**Published:** 2021-08-23

**Authors:** Stefania Costi, Roberto Tonelli, Antonio Brogneri, Fabio Florini, Nicolò Tilocca, Massimo Vicentini, Serena Baroncini, Massimo Cerulli, Enrico Clini

**Affiliations:** 1Surgical, Medical and Dental Department of Morphological Sciences Related to Transplant, Oncology and Regenerative Medicine CHIMOMO, University of Modena and Reggio Emilia, Via del Pozzo n.71, 41124 Modena, Italy; stefania.costi@unimore.it; 2Scientific Directorate, Azienda Unità Sanitaria Locale-IRCCS di Reggio Emilia, Viale Umberto I n.50, 42123 Reggio Emilia, Italy; 3Clinical and Experimental Medicine PhD Program, University of Modena Reggio Emilia, Via del Pozzo n.71, 41124 Modena, Italy; 4Respiratory Diseases Unit, Department of Medical and Surgical Sciences SMECHIMAI, University of Modena and Reggio Emilia, University Hospital of Modena, Via del Pozzo n.71, 41124 Modena, Italy; enrico.clini@unimore.it; 5Respiratory Rehabilitation of Ospedale Villa Pineta-KOS Group, Via Gaiato n. 127, Pavullo nel Frignano, 41026 Modena, Italy; antonio.brogneri@libero.it (A.B.); florini.fabio@villapineta.it (F.F.); tiloccatilocca@gmail.com (N.T.); massimo.cerulli@villapineta.it (M.C.); 6Epidemiology Unit, Azienda Unità Sanitaria Locale-IRCCS di Reggio Emilia, Viale Amendola n.2, 42123 Reggio Emilia, Italy; massimo.vicentini@ausl.re.it; 7School of Respiratory Medicine, University of Modena Reggio Emilia, Via del Pozzo n.71, 41124 Modena, Italy; serena.baroncini@gmail.com

**Keywords:** cardiac rehabilitation, cardiovascular diseases, treatment outcome, comorbidity, elderly, exercise

## Abstract

Aims: Cardiac rehabilitation (CR) has proven to be effective and beneficial in middle-aged and older patients. However, solid data in large cohorts of elderly individuals are yet to be explored. This retrospective study investigated the general characteristics, outcomes, and the level of response of patients referred to CR over 13 consecutive years. Methods: We reviewed the medical records of patients admitted to Villa Pineta Rehabilitation Hospital for exercise-based CR from 2006 to 2018. The patients’ baseline characteristics and changes following CR in an upper-limb weightlifting test (ULW), 30-s sit-to-stand test (30STS), and the 6-min walking test (6MWT) with associated Borg-related dyspnea (D) and fatigue (F) were collected. We also calculated the number of individuals that reached the minimal clinically relevant change (MCRC) following CR for each outcome. Results: One thousand five hundred and fifty-one patients (70.2 ± 9.7 years, 66% men) with complete datasets were included in the analysis. Coronary artery bypass graft and cardiac valve replacement surgery were the most frequent surgical procedures leading to CR referral (41.1% and 35.8%, respectively). The patients’ age (*p* = 0.03), number of total comorbidities (*p* < 0.0001), and post-surgical complications (*p* = 0.02) significantly increased over time. In contrast, the average absolute changes in ULW, 30STS, and 6MWT with associated D and F, and the proportion of patients that reached their respective MCRC, remained constant over the same period. Conclusion: The patients admitted to exercise-based CR were older and had more comorbidities and complications over time. The outcomes, however, were not influenced in terms of the absolute change or clinically meaningful response.

## 1. Introduction

Cardiovascular diseases (CVDs) are the leading cause of morbidity and mortality in Western countries, with a high socioeconomic burden [1].

Cardiac rehabilitation (CR) is a multidimensional secondary prevention program that has become a standardized component of CVD management, since CR has proven to be beneficial for mobility, muscle strength and mass, physical performance, social participation, and the mood of patients with CVDs [1,2]. 

Although CR originated as an exercise training program addressing middle-aged individuals affected by coronary heart disease, it has evolved into a comprehensive program promoting an active and healthier lifestyle for a broad range of CVDs (coronary heart disease, heart failure, valvular heart disease, etc.) through education, diet, and risk factor reduction [3]. Thus, CR is currently recommended for individuals affected by different CVDs, with increasing evidence of its beneficial effects in older populations [2]. 

The advances in treatment have led to the chronification of CVDs, and the rationale for recommending CR to the expanding senior population is sound, as most of the risk factors for CVDs are age-related. Nonetheless, very few older individuals participate in CR [4], and compelling evidence has pointed out a lack of healthcare professional referral of all patients who could benefit from it [5]. CR could lessen the impact of CVDs on the physical performance and quality of life in the elderly, who can especially be affected [6]. However, CR in the elderly must be adapted to several geriatric variables, such as frailty, cardiovascular and non-cardiovascular multimorbidity, polypharmacy, physical deconditioning, and declining cognition, among others. Thus, CVD management in this population always requires a fair balance between costs and benefits; the focus should be on the patient’s needs, avoiding burdensome treatments in the face of minimal benefits. Hence, collecting further evidence on the effectiveness of CR on clinically relevant outcomes in eligible patients is recommended, particularly in the elderly population, in order to increase the referral and participation rates [7].

The aim of this retrospective study was to investigate the demographic and the clinical characteristics and outcomes of patients referred to CR over 13 consecutive years. If changes occurred over time, we also examined the level of response to CR (i.e., the achievement of clinically relevant gains in the outcomes collected) in the same period.

## 2. Methods

### 2.1. Study Design and Data Sources

This retrospective study investigated a cohort of patients admitted for inpatient CR after cardiac surgery over a 13-year period (2006–2018). 

We retrieved data from the medical records database of the Villa Pineta Rehabilitation Hospital-KOS Group (Gaiato di Pavullo, Modena, Italy). The database contained information on inpatient care, including demographic information, diagnoses, surgical procedures, CR program, and the assessments performed. 

Consent to collect personal data and to use it in observational studies was obtained from all patients on admission. Moreover, as per the protocol approved by the local ethics committee (AOU: 0011677/19, date 18 April 2019), specific consent to use the data for the purposes of this study was obtained from patients who were still in follow-up at the time of data collection. 

### 2.2. Cardiac Rehabilitation Program

On admission, all patients underwent a thorough assessment of their clinical, physical, and psychosocial characteristics by the multidisciplinary team, which included cardiologists, pneumologists, physiotherapists, nurses, dieticians, and psychologists. After a team discussion, a personalized exercise-based rehabilitation program targeting patient-specific objectives was developed by the physiotherapists and cardiologists. The prerequisite for this individualized program was to target the patient’s maximal tolerated activity to achieve the best results (outcome measures).

The CR program included exercise training, physical activity and nutritional counselling, weight control, psychosocial status management, medication rationalization, and strategies to keep the CVD risk factors under control, as recommended [1]. 

The muscle training program consisted of intensive daily 2-h supervised exercise (generally once in the morning and once in the afternoon). The program included active mobilization of the shoulder girdle; upper and lower limb muscle stretching; and aerobic exercises (e.g., stationary bicycle, treadmill, etc.) at increasing loads.

CR also included the treatment of surgical wounds and chest physiotherapy to assist and promote the clearance of excessive bronchial secretions.

### 2.3. Data Collection and Outcome Assessment

Data regarding age; sex; self-reported comorbidities (as assessed by the Charlson Comorbidity Index); type of surgery; duration of inpatient stay; the number of CR sessions performed; and the number of complications post-surgery (i.e., infections, bleeding, etc.) were collected from the medical records database. The assessments of the CR outcomes (i.e., the measures of functional muscle strength and walking exercise capacity) were also compared at the baseline (T0) and after CR (T1) by means of the following pre-defined measures: upper-limb weightlifting test (ULW), a 30-s sit-to-stand test (30STS), a 6-min walking test (6MWT), and the associated level of dyspnea (D) and fatigue (F) measured using the Borg scale [8]. The 30STS and the 6MWT were conducted according to the standardized recommendations [9,10]. The ULW test consisted of recording the number of full flexions and extensions of the elbows that male and female patients could perform while lifting weights of 2 kg and 1 kg, respectively. The patients were seated on the same chair used for the 30STS, with their upper limbs adducted, fully extended, and extra-rotated.

The minimal clinically relevant change (MCRC) following CR was also individually calculated for each outcome as follows: ≥1 repetition for ULW [11,12], >2 repetitions for the 30STS [13,14], +30 m for 6MWD [15], and −1 point for BORG D-F [16,17]. 

### 2.4. Data Analysis

The statistical packages SPSS version 25.0 (IBM Corp., New York, NY, USA) [18], GraphPad Prism 8.0 (GraphPad Software, Inc., La Jolla, CA, USA) [19], and Jamovi 1.2.27 (Jamovi Project, Sydney, NSW, Australia) [20] were used for the statistical analyses. Descriptive statistics reporting the numbers and percentages for the dichotomous variables and means (±standard deviations (SD)) and medians (interquartile ranges (IQR)) for the continuous variables were applied to describe the data year by year. The average values (±SD) or medians (IQR) for the functional outcomes of the CR were described at the baseline (T0) and after the completion of CR (T1), as well as their changes. Before the analysis, the data were checked with a Bartlett’s test for equal distribution of the variances. A linear mixed effects model was then built to assess the changes in the average values of the baseline characteristics across the years, considering both patients and years as random effects to avoid an estimation bias. The absolute numbers and percentages of patients who reached the MCRC after the CR for each outcome were reported. Three categories of patients with different response profiles were generated as follows and then considered for the analysis: (a) high responders (five outcomes reaching the MCRC at T1), (b) moderate responders (three to four outcomes reaching the MCRC), and (c) low responders (up to two outcomes reaching the MCRC). A chi-square test was performed to test the changes in the proportions within the time (years)-dependent groups. A two-sided test of less than 0.05 was considered statistically significant.

## 3. Results

From January 2006 to December 2018, two thousand and five individuals underwent CR at our hospital, of whom 364 were not deemed eligible, as they had not undergone recent surgery. We therefore reviewed 1641 medical records; of these, 90 were excluded for the reasons illustrated in Figure 1.

A total number of 1551 patients with complete pre-to-post data were included in the study, and their main characteristics by year are shown in Table 1. Overall, the average patient age was 70.2 years (±9.7), and 66% were men (*n* = 1018). Coronary artery bypass graft (CABG) and cardiac valve replacement surgery (CVS) were the most represented surgical procedures (41.1% and 35.8%, respectively), followed by plastic surgery of the mitral valve (10.3%); the remaining patients reported mixed conditions. The median index score of the chronic comorbidities was 2 (IQR 1;4), with hypertension (58.6%) and cardiac arrhythmias (43.2%) being the most frequent. The complications post-surgery were relatively few (see the details in Table 1). During their hospital stay (16 ± 3.6 days), the patients attended 13.2 (±2.7) sessions of CR. 

Over the 13-year timespan, the patient age (*p* = 0.03), number of total comorbidities (*p* < 0.0001), and clinical complications post-surgery (*p* = 0.02) significantly increased, whereas the length of hospital stay and the number of CR sessions attended decreased (*p* < 0.0001 for both).

Table 2 reports the CR outcomes (see the Methods section) both at T0 and T1 and displays the pre-to-post changes following the CR, which were all statistically significant (*p* < 0.001) and which indicated a good response to the CR. The average changes were not significantly different over the years for all the predetermined outcomes. The number of individuals who reached the MCRC for each CR outcome during the period examined is reported in Table 3; only Borg F showed a significant reduction in the proportion of patients reaching the MCRC (*p* < 0.001) over the years.

According to the categories of the response to CR (see the Methods section), the majority of the cases were high (24.4%; *n* = 379) or moderate (71.3%, *n* =1106) responders, whereas only a very few (4.3%, *n* = 66) were poor responders. Figure 2 shows the distribution of the categories of response across the timespan, revealing an increase in moderate responders at the expense of the high-response category within the same timeframe (*p* = 0.02)

## 4. Discussion

This retrospective study reports unique information on the outcomes following a standard exercise-based CR in a very large population of patients over a considerable period of time. We observed that patients’ age and clinical characteristics significantly changed over the 13 years examined. Notwithstanding this, the predetermined outcomes were not influenced in terms of the absolute change and in terms of the number of patients reaching the respective MCRC. Of note, the categories of response showed a progressive percentage increase in the “moderate responders”, whilst the proportion of “low responders” remained stable.

Overall, we believe that the main finding of this study is that it confirmed a good quantitative response to CR [2], irrespective of the clinical characteristics of the patients admitted. Cardiac patient referrals to rehabilitation may differ and/or change over time for several reasons. In our center, located in Northeastern Italy, referrals are for post-surgical patients, who represent the core business and interest of the local (regional) stakeholders. Across the 13 years studied, older patients with more comorbidities were referred for surgery, since medical care has improved [21] and minimally invasive techniques are more frequently used in younger patients at a higher risk [22]. Nonetheless, comorbidities in chronic and disabled individuals do not preclude access to rehabilitation and its benefits [23].

Very recently, a systematic review including six trials and 364 participants confirmed that exercise-based CR is likely to improve specific outcomes in patients referred after heart valve surgery [24], although the effects in terms of mortality, hospitalization, and health-related quality of life are still uncertain and/or partly unexamined. Our study therefore further supports the indication for CR after cardiac surgery (>70% of patients/year) irrespective of age and comorbidities, based on both the absolute change in CR outcomes (see Table 2) and the proportion of patients reaching the clinically meaningful criteria for improvement (see Table 3). Of note, the MCRCs did not substantially change over time in all the CR outcomes, except for Borg-F, as assessed during the exercise tolerance test. It is difficult to explain this result; it may be associated with the changes in the patients’ characteristics observed over time, with a progressive increase in the referral to CR of older patients with more comorbidities less responsive to perceived fatigue [25]. 

Thus, to date, CR after surgery still remains a valid therapeutic, nonpharmacological opportunity for eligible candidates. Further, it appears much better than preoperative physiotherapy, which, according to the literature, still shows conflicting results [26,27]. 

It is noteworthy that the quantitative gains showed by the cohort examined were maintained by reducing the number of days spent in hospital and (slightly) the mean number of CR sessions attended per patient, thus making a better cost-effective intervention possible in more recent years. The considerable timespan observed might have indeed improved the staff’s ability to use resources better and/or may demonstrate an optimization of the learning curve of healthcare professionals working with people with multiple comorbidities [28]; however, a formal cost-effective analysis was not possible with this retrospective design, and it was beyond the scope of the study.

Finally, we were also able to demonstrate that the qualitative response (i.e., the response categories as described in the Methods section) did not substantially change over time, although the proportion of “moderate responders” increased compared to the “high responders” (see Figure 2). It is questionable whether this variation had any significant implication on the effectiveness of the CR program (which remained constant over time). Nonetheless, over the years, the number of poor responders remained stable and very low (less than 5%) across the treated population. 

Notwithstanding the positive findings observed, this retrospective study has obvious limitations. Firstly, this retrospective study provided information that needs to be confirmed prospectively. Secondly, due to the quite long time period examined, the patients’ post-surgery recovery pathway could have been affected both by the evolution in surgical procedures and by an increase in the staff’s experience. Thirdly, as the study was conducted in a single center, with a unique rehabilitation approach, our results cannot be generalized or considered as valid for any candidate to CR. Nevertheless, to the best of our knowledge, this study is the first to show a composite treatment response evaluation after CR. Previous reports in the rehabilitation of chronic respiratory patients have shown that the key performance measures to evaluate the effectiveness of programs had to be selected within multiple domains, including those related to what a patient perceives in his/her daily life [29]. Despite the fact that, in this study, we did not collect a wide range of measures to assess the patients’ outcomes, we do believe that both the selected performance measures [1,2] and the composite response to these performances might have reflected the benefits obtained in those patients’ everyday lives.

## 5. Conclusions

In conclusion, this retrospective analysis of a very large population confirms the benefits of CR even in older patients with comorbidities undergoing cardiac surgery and opens a window into the area of composite responses in this specific field of rehabilitation. 

## Figures and Tables

**Figure 1 ijerph-18-08871-f001:**
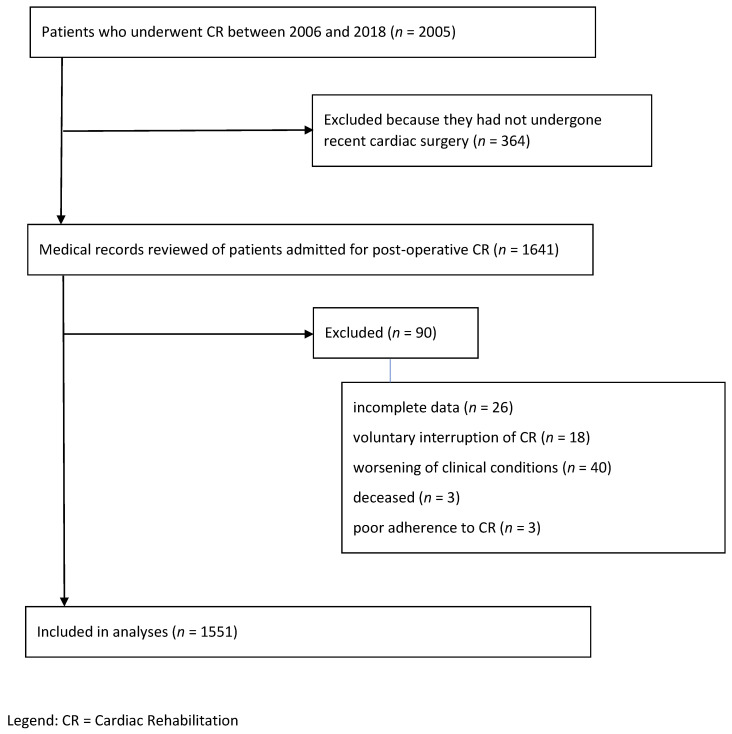
A flowchart of the patients’ records reviewed for eligibility.

**Figure 2 ijerph-18-08871-f002:**
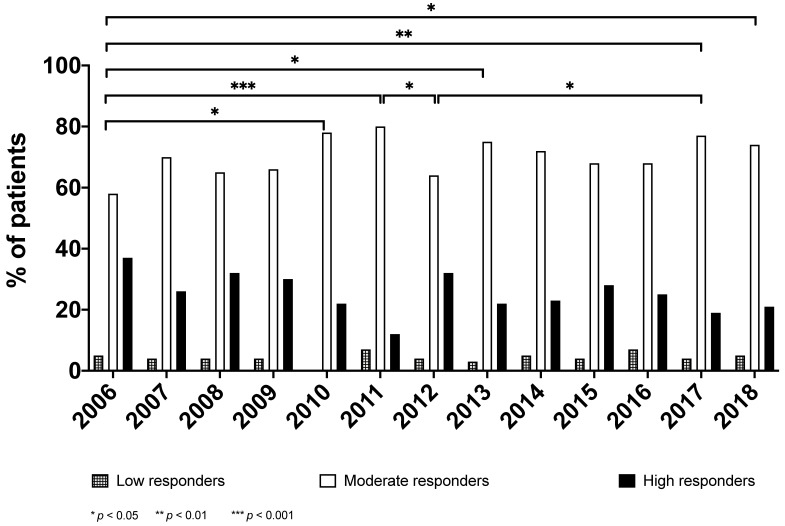
Clusters of responses across 13 years.

**Table 1 ijerph-18-08871-t001:** The characteristics of the study cohort by year.

Year	Pt.	F	M	Age	Main Diagnosis	Comorbid.	Complic.	CR Sessions	Hospital Stay
	*n*	*n* (%)	*n* (%)	Mean ± SD	Freq. (%)	Median IQR (25;75)	Median IQR (25;75)	Mean ± SD	Mean ± SD
2006	65	22 (34)	43 (66)	69.7 ± 10.1	CABG (54%)	1 (1;3)	0 (0;1)	13.8 ± 2.9	16.5 ± 3.3
2007	94	24 (26)	70 (75)	68.0 ± 10.1	CABG (44%)	2 (1;3)	0 (0;1)	14.1 ± 3.3	17.0 ± 4.0
2008	79	24 (30)	55 (70)	68.6 ± 9.3	CABG (41%)	1 (0;3)	0 (0;1)	14.8 ± 3.2	18.1 ± 4.6
2009	112	38 (34)	74 (66)	69.5 ± 10.1	CABG (47%)	2 (1;3)	0 (0;1)	13.4 ± 3.5	16.5 ± 4.4
2010	83	30 (36)	53 (64)	70.2 ± 9.7	CABG (46%)	2 (1;3)	1 (0;1)	13.6 ± 3.6	16.5 ± 4.8
2011	121	38 (31)	83 (69)	70.0 ± 10.5	CVS (38%)	2 (1;4)	0 (0;1)	13.3 ± 2.8	16.3 ± 3.9
2012	168	67 (40)	101 (60)	70.0 ± 9.5	CVS (41%)	2 (1;4)	0 (0;1)	12.8 ± 2.7	15.9 ± 3.7
2013	193	60 (31)	133 (69)	70.9 ± 10.2	CVS (40%)	2 (1;3)	0 (0;1)	12.7 ± 2.1	15.4 ± 2.9
2014	87	40 (46)	47 (54)	69.8 ± 11.6	CVS (45%)	3 (2;4)	1 (0;1)	12.9 ± 2.4	15.9 ± 2.9
2015	100	33 (33)	67 (67)	69.6 ± 9.5	CABG (41%)	2 (1;3)	0 (0;1)	13.2 ± 3.6	16.1 ± 5.3
2016	104	48 (46)	56 (54)	70.9 ± 9.3	CVS (47%)	2 (1;3)	0 (0;1)	12.7 ± 1.6	15.3 ± 1.9
2017	197	65 (33)	132 (67)	71.4 ± 8.4	CABG (44%)	3 (2;4)	0 (0;1)	12.9 ± 1.6	15.5 ± 2.0
2018	148	44 (30)	104 (70)	71.1 ± 9.6	CABG (43%)	3 (2;4)	1 (0;1)	12.8 ± 1.8	15.4 ± 2.7
Total	1551	533 (34)	1018 (66)	70.2 ± 9.7	CABG (41%)	2 (1;4)	0 (0;1)	13.2 ± 2.7	16.0 ± 3.6

Note: Pt., Patients; F, Female; M, Male; Comorbid., Comorbidities; Complic., Complications; CR, Cardiac Rehabilitation; *n*, number; SD, Standard Deviation; Freq., Frequency; IQR, Interquartile Range; CABG, Coronary Artery Bypass Graft; CVS, Cardiac Valve replacement Surgery.

**Table 2 ijerph-18-08871-t002:** Outcome changes over the CR course and during the considered timespan.

Year	ULW			30STS			6MWD			Borg D			Borg F		
	T0	T1	Δ	T0	T1	Δ	T0	T1	Δ	T0	T1	Δ	T0	T1	Δ
	Mean ± SD			Mean ± SD			Mean ± SD			Median IQR (25;75)			Median IQR (25;75)		
2006	11.9 ± 6.9	18.5 ± 6.7	6.5 ± 4.7	6.8 ± 5.9	10.3 ± 7.0	3.5 ± 3.9	239 ± 161	354 ± 142	115 ± 93	6 (5;7)	2 (2;4)	−3 (−4;−2)	6 (5;7)	3 (2;4)	−3 (−4;−2)
2007	13.4 ± 6.2	19.9 ± 6.5	6.4 ± 5.1	7.5 ± 5.2	10.9 ± 6.2	3.2 ± 3.0	242 ± 132	336 ± 131	94 ± 74	5 (4;7)	2 (1;4)	−3 (−4;−2)	5 (4;7)	2 (1;4)	−3 (−4;−2)
2008	14.3 ± 7.0	19.8 ± 6.9	5.0 ± 4.9	7.1 ± 6.1	10.8 ± 7.0	3.5 ± 3.0	235 ± 139	342 ± 131	107 ± 71	6 (5;7)	2 (1;4)	−3 (−4;−2)	6 (5;7)	2 (1;3)	−3 (−5;−2)
2009	14.4 ± 6.7	20.5 ± 6.4	5.7 ± 5.1	8.0 ± 5.4	12.2 ± 6.2	3.4 ± 3.4	231 ± 131	342 ± 127	110 ± 72	6 (4;7)	2 (2;3)	−3 (−4;−2)	6 (4;8)	3 (2;4)	−3 (−4;−2)
2010	19.4 ± 6.6	25.2 ± 8.0	4.7 ± 6.9	9.5 ± 4.6	13.9 ± 5.8	3.2 ± 3.3	243 ± 131	356 ± 118	114 ± 63	5 (3;6)	2 (1;3)	−2 (−4;−1)	5 (3;6)	2 (1;3)	−3 (−4;−2)
2011	19.7 ± 8.2	25.3 ± 8.1	4.8 ± 6.2	9.4 ± 6.1	13.3 ± 5.8	2.9 ± 3.3	263 ± 137	359 ± 133	96 ± 62	4 (3;6)	2 (2;3)	−2 (−3;−1)	4 (3;6)	2 (1;3)	−2 (−3;−1)
2012	18.7 ± 6.4	25.0 ± 6.6	5.4 ± 5.0	8.7 ± 4.5	13.1 ± 5.2	3.3 ± 3.1	237 ± 120	346 ± 119	109 ± 67	5 (3;7)	2 (1;3)	−2 (−4;−1)	4 (3;6)	2 (1;3)	−2 (−4;−1)
2013	19.6 ± 6.1	25.2 ± 5.7	4.9 ± 4.3	9.6 ± 4.5	13.4 ± 5.1	3.1 ± 2.9	259 ± 118	357 ± 126	99 ± 63	4 (3;5)	2 (1;2)	−2 (−3;−1)	4 (2;6)	2 (1;3)	−2 (−3;−1)
2014	20.1 ± 7.7	26.9 ± 10.8	6.2 ± 8.5	8.8 ± 4.9	13.4 ± 5.0	3.5 ± 3.5	241 ± 122	356 ± 117	115 ± 64	5 (3;8)	2 (1;3)	−3 (−4;−2)	4 (3;7)	2 (1;4)	−2 (−4;−1)
2015	17.7 ± 6.4	24.9 ± 7.6	6.7 ± 5.7	9.5 ± 4.8	13.6 ± 5.2	3.4 ± 3.2	271 ± 118	389 ± 124	118 ± 77	4 (3;7)	2 (1;3)	−2 (−4;−1)	4 (2;7)	2 (0;3)	−2 (−4;−1)
2016	18.3 ± 6.8	24.9 ± 7.2	6.0 ± 6.2	8.6 ± 5.3	12.9 ± 6.2	3.6 ± 4.3	270 ± 119	379 ± 122	110 ± 70	4 (3;5)	2 (1;3)	−2 (−3;−1)	4 (3;5)	2 (1;2)	−2 (−3;−1)
2017	19.3 ± 6.7	25.4 ± 7.1	5.6 ± 5.2	8.3 ± 5.2	12.5 ± 5.0	3.6 ± 3.1	271 ± 108	378 ± 110	107 ± 61	5 (3;6)	2 (1;3)	−3 (−4;−1)	4 (2;6)	2 (1;3)	−2 (−4; 0)
2018	17.9 ± 6.2	23.4 ± 6.2	5.5 ± 5.0	8.4 ± 4.5	12.0 ± 4.2	3.5 ± 2.4	267 ± 105	376 ± 122	109 ± 61	4 (3;6)	2 (1;3)	−3 (−4;−1)	4 (2;6)	2 (1;3)	−2 (−4;−1)
Total	17.7 ± 7.1	23.8 ± 7.5	5.6 ± 4.9	8.6 ± 5.2	12.6 ± 5.6	3.4 ± 4.2	254 ± 124	361 ± 124	107 ± 68	5 (3;7)	2 (1;3)	−3 (−4;−1)	5 (3;7)	2 (1;3)	−2 (−4;−1)

Note: ULT, Upper-Limb Weightlifting; 30STS, 30-sec Sit-To-Stand test; 6MWD, 6-Min Walked Distance; D, Dyspnea; F, Fatigue; T0, baseline assessment; T1, follow-up; Δ, changes after cardiac rehabilitation; SD, Standard Deviation; IQR, Interquartile Range.

**Table 3 ijerph-18-08871-t003:** Patients reaching the MCRC in each outcome following the CR and during the considered timespan.

Year	ΔULW		Δ30STS		Δ6MWD		ΔBorg D		ΔBorg F	
	*n*	%	*n*	%	*n*	%	*n*	%	*n*	%
2006	42	64.6	37	56.9	58	89.2	58	89.2	56	86.2
2007	54	57.4	47	50.0	80	85.1	85	90.4	85	90.4
2008	42	53.2	45	57.0	68	86.1	70	88.6	74	93.7
2009	60	53.6	65	58.0	103	92.0	100	89.3	101	90.2
2010	36	43.4	42	50.6	80	96.4	72	86.7	75	90.4
2011	52	43.0	55	45.5	106	87.6	94	77.7	101	83.5
2012	88	52.4	92	54.8	152	90.5	145	86.3	143	85.1
2013	90	46.6	112	58.0	171	88.6	159	82.4	160	82.9
2014	45	51.7	48	55.2	80	92.0	75	86.2	65	74.7
2015	58	58.0	56	56.0	94	94.0	87	87.0	77	77.0
2016	55	52.9	62	59.6	95	91.3	83	79.8	81	77.9
2017	104	52.8	121	61.4	186	94.4	162	82.2	135	68.5
2018	73	49.3	84	56.8	140	94.6	123	83.1	115	77.7
Total	799	51.5	866	55.8	1413	91.1	1313	84.7	1268	81.8

Note: ULT, Upper-Limb Weightlifting; 30STS, 30-s Sit-To-Stand test; 6MWD, 6-Min Walked Distance; D, Dyspnea; F, Fatigue.

## Data Availability

The data underlying this article cannot be shared publicly due to the privacy of the individuals that contributed to the study with their data. The data can be shared on reasonable request to the corresponding author.

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
