# Peer review of "Changes in Clinical Characteristics and Outcomes of Patients Admitted to Inpatient Cardiac Rehabilitation"

_ijerph, 2021, doi:10.3390/ijerph18168871_

Round 1

Reviewer 1 Report

Abstract

Line 23-24 “….solid data in large cohorts of aged individuals are still to come”. “Still to come” is poor English, and I would suggest replacing this with something like “are yet to be explored” or similar.

Line 26 – the address of the hospital does not need to be mentioned in the abstract

Line 35 – “Differently” is poor English and would perhaps be better replaced with “In contrast”

All – there is a duplication of referencing styles throughout the document (Harvard and Vancouver styles are used simultaneously). I would recommend using Vancouver only, in order to cut down on the overall word count.

Introduction

Line 58 – “Besides” is a bit colloquial – something like “Moreover” would be more appropriate.

Line 61 – The colon following ‘benefits’ should be replaced with a semicolon.

Line 66 – ‘Therefore’ is unnecessary here.

Lines 64-68 – I felt that the aims paragraph was incongruent with the rest of the introduction section. The stated aim of the study is to look at whether the clinical characteristics and demographics of CR patients had changed over time, but there was no mention of this temporal dimension elsewhere in the introduction. This made me question the relevance (and value) in looking at these clinical/demographic changes over time. I also had trouble understanding why more emphasis was placed on changes in patients’ demographics/clinical characteristics, as opposed to those patients’ outcomes. Surely when evaluating the worth of a given service, the outcomes of its patients is of more interest than their clinical/demographic characteristics? I’m sure that the authors had their reasons for doing so, but these need to be made much clearer to the reader.

Methods

Line 75 – the apostrophe in ‘inpatient’s’ is unnecessary. ‘Inpatient’ is fine on its own.

Line 80 – there is no mention of which clinical groups comprise the multidisciplinary team

Line 82 – which of the above groups actually create the personalised plans?

Line 93 – Critically, there is no mention of how ‘Combordities’ are defined, and whether the same measure applied across the whole cohort – despite comorbidities being mentioned elsewhere in the manuscript. For example, was a Charlson Comobidity Index score or Elixhauser score used? On a similar note, there are bound to be other variables which would have influenced patients’ outcomes (e.g. smoking status is one). The value in recording data in relation to patients’ comorbidities is also limited because these factors were not statistically accounted for. The study design would have benefited greatly from the use of a regression model of some description, which could have accounted for patient characteristics when assessing whether there had been an improvement in outcomes. ‘Complications’ are another factor which is mentioned in the results section but not in the methods – how was this defined?

Results

The study’s tables were poorly made, with several redundant columns and wrapped text which is difficult to read. Table 2 was particularly difficult to read, given that the ‘title’ rows were on a separate page to the rest of the table.

Discussion

The study’s conclusions are not supported because of how ill-defined ‘clinical complexity’ is. There is also a significant risk that perceived improvements over time were the result of residual confounding, given that the abovementioned ‘complexity’ was not statistically accounted for.

Author Response

Reviewer 1

Abstract

Line 23-24 “….solid data in large cohorts of aged individuals are still to come”. “Still to come” is poor English, and I would suggest replacing this with something like “are yet to be explored” or similar.

Response: Thank you for all the language improvements that have been suggested. These have been made (Abstract line 25).

Line 26 – the address of the hospital does not need to be mentioned in the abstract

Response: It has been deleted (Abstract line 28).

Line 35 – “Differently” is poor English and would perhaps be better replaced with “In contrast”

Response: The change suggested has been made (Abstract line 38).

All – there is a duplication of referencing styles throughout the document (Harvard and Vancouver styles are used simultaneously). I would recommend using Vancouver only, in order to cut down on the overall word count.

Response: I apologize for the mistake. We have chosen the Vancouver style.

Introduction

Line 58 – “Besides” is a bit colloquial – something like “Moreover” would be more appropriate.

Response: This change has been made (Introduction line 60).

Line 61 – The colon following ‘benefits’ should be replaced with a semicolon.

Response: Thank you (Introduction line 63).

Line 66 – ‘Therefore’ is unnecessary here.

Response: Thank you (Introduction line 68).

Lines 64-68 – I felt that the aims paragraph was incongruent with the rest of the introduction section. The stated aim of the study is to look at whether the clinical characteristics and demographics of CR patients had changed over time, but there was no mention of this temporal dimension elsewhere in the introduction. This made me question the relevance (and value) in looking at these clinical/demographic changes over time. I also had trouble understanding why more emphasis was placed on changes in patients’ demographics/clinical characteristics, as opposed to those patients’ outcomes. Surely when evaluating the worth of a given service, the outcomes of its patients is of more interest than their clinical/demographic characteristics? I’m sure that the authors had their reasons for doing so, but these need to be made much clearer to the reader.

Response: We thank the reviewer for this useful comment. We agree that the aim is a little bit misleading and have therefore, based on the suggestion, rephrased it in this revised version. We believe we have now clarified that the aims were to look both at patients’ clinical/demographic characteristics and at CR outcomes over the time span examined.

Methods

Line 75 – the apostrophe in ‘inpatient’s’ is unnecessary. ‘Inpatient’ is fine on its own.

Response: This has been corrected (Methods/2.1. Study design and data sources, line 77).

Line 80 – there is no mention of which clinical groups comprise the multidisciplinary team

Response: The multidisciplinary team included cardiologists, pneumologists, physiotherapists, nurses, dieticians and psychologists. We have specified this in the revised version (Methods/2.2. Cardiac rehabilitation program, lines 87-88).

Line 82 – which of the above groups actually create the personalised plans?

Response:  The comprehensive CR program, which includes exercise training, physical activity and nutritional counselling, weight control, psychosocial status management, medication rationalization and strategies to keep CVD risk factors under control, is developed by the whole team. The personalized exercise-based rehabilitation program targeting the patient-specific objectives was developed by physiotherapists and cardiologists, after team discussion. We have specified this in the revised version (Methods/2.2. Cardiac rehabilitation program, lines 90).

Line 93 – Critically, there is no mention of how ‘Combordities’ are defined, and whether the same measure applied across the whole cohort – despite comorbidities being mentioned elsewhere in the manuscript. For example, was a Charlson Comobidity Index score or Elixhauser score used? On a similar note, there are bound to be other variables which would have influenced patients’ outcomes (e.g. smoking status is one). The value in recording data in relation to patients’ comorbidities is also limited because these factors were not statistically accounted for. The study design would have benefited greatly from the use of a regression model of some description, which could have accounted for patient characteristics when assessing whether there had been an improvement in outcomes. ‘Complications’ are another factor which is mentioned in the results section but not in the methods – how was this defined?

 Response: We thank the reviewer for this useful comment. Comorbidities were based on a self-reported measure, namely the CCI score, retrieved from each patient’s record. Complications were any event (e.g., bleeding, infection, etc.) following surgery and during the CR period. This has been more clearly explained in the Methods section of the new version. Data are reported per year and analyzed throughout for comparison (see Table 1). Predictive analysis to look at each patient’s risk factors that may have modified the CR outcome(s) was not considered in the present study.

Results

The study’s tables were poorly made, with several redundant columns and wrapped text which is difficult to read. Table 2 was particularly difficult to read, given that the ‘title’ rows were on a separate page to the rest of the table.

Response: We have improved the format of Table 1 and have completely reorganized the content of Table 2.

Discussion

The study’s conclusions are not supported because of how ill-defined ‘clinical complexity’ is. There is also a significant risk that perceived improvements over time were the result of residual confounding, given that the abovementioned ‘complexity’ was not statistically accounted for.

Response: We thank the reviewer for this useful comment. We have eliminated the term complexity from the present version, substituting it with “number. of comorbidities” (which reflects a patient’s characteristic and which refers to what has been measured retrospectively). This has been changed throughout the manuscript, with the Discussion/Conclusions paragraphs modified accordingly.

Reviewer 2 Report

  1. What is the relevance for measure ULW, 30STS, 6MWD, and Borg scale for the present protocol? Some information should be integrated into the introduction section to clarify the above. On the other hand, what other parameters could be useful for this work?
  2. An observation in figure 2 is that the high responder population decreases over time (2006 vs 2018), could the authors discuss this?
  3. If the CR program was personalized for each patient, the above could have affected the final results?
  4. Choose a style for references in the text: superscript number or (name, year)
  5. The informed consent for the patients was obtained before including them in the study? Or after the approbation of the protocol for the ethical committee? Please, indicate.
  6. Adjust the legends of Table 1 in a line for instance; mean, frequent, etc.
  7. Include the symbol ± after the mean. I suppose that SD refers to standard deviation.
  8. In table 2, it is clear that measurements at T1 are increased compared against T0, indicating a better response induced by CR, but, this increase is significant, could the authors include the p-value?
  9. Improve the format of the tables included in the manuscript.

Author Response

Reviewer 2

  1. What is the relevance for measure ULW, 30STS, 6MWD, and Borg scale for the present protocol? Some information should be integrated into the introduction section to clarify the above. On the other hand, what other parameters could be useful for this work?

Response: These measures are some of the standard outcomes used in CR to assess treatment efficacy. In the original version of this manuscript, we commented that our predetermined measures were not the only ones and/or the most representative (see Discussion, line 280-286). However, we are confident that the selected performance measures are responsive to CR (cfr. Ref. # 1,2). We have slightly modified the Discussion in the revised version accordingly.

  1. An observation in figure 2 is that the high responder population decreases over time (2006 vs 2018), could the authors discuss this?

Response: Figure 2 illustrates the change in the categories of response, clearly showing that the number of high responders decreased slightly over time, and that the increase in the number of moderate responders mirrored this change. It is questionable whether this change had any significant implication on the effectiveness of the CR program (which remained constant over time), since the number of poor responders remained stable and very low across the treated population. A brief comment has been added to the Discussion of the revised version of our manuscript.

  1. If the CR program was personalized for each patient, the above could have affected the final results?

Response: As rehabilitation programs are always personalized and adapted to the patient’s characteristics and abilities (i.e., intensity progression over the programmed sessions, etc.), the prerequisite for this individualized program was to target the patient’s maximum tolerated activity to achieve the best result (outcome measures). A brief comment on this has been added to the Methods section of the revised version.

  1. Choose a style for references in the text: superscript number or (name, year)

Response: I apologize for the mistake. We have chosen the Vancouver style.

  1. The informed consent for the patients was obtained before including them in the study? Or after the approvation of the protocol for the ethical committee? Please, indicate.

Response: This is a retrospective study which collected data of patients admitted for CR over 13 years. Although at the Villa Pineta Hospital a generic consent to collect personal data and to use them in observational studies is usually requested from all patients on admission, the ethics committee that approved this study also asked us to collect a new consent specific to this study from all the patients who, at the time of the study, were still in follow up at the hospital. We have now explained how we collected the patients’ informed consent in the Methods section of the new version (lines 79-84).

  1. Adjust the legends of Table 1 in a line for instance; mean, frequent, etc.

Response: We have changed the format of Table 1 to improve its readability and comprehensibility.

  1. Include the symbol ± after the mean. I suppose that SD refers to standard deviation.
  1. Response: We have added this to the tables and have also included a complete legend note to improve readability and comprehensibility.

  1. In table 2, it is clear that measurements at T1 are increased compared against T0, indicating a better response induced by CR, but, this increase is significant, could the authors include the p-value?

In fact, all the comparisons were significant at p < 0.001; we have included this information in the text (Results, page 5, lines 179).

  1. Improve the format of the tables included in the manuscript.

Response: We have improved the format of Table 1 and have completely reorganized the content of Table 2.

Round 2

Reviewer 1 Report

Thanks for your edits. 

I refer to my initial comment "Lines 64-68 – I felt that the aims paragraph was incongruent with the rest of the introduction section".

I think there still needs to be a stronger rationale for looking at changes in patient demographics/outcome over time. Furthermore, if you're interested in the efficacy of CR in elderly patients, why not look at this group in isolation? I fail to see the value in assessing whether the average patient was older at the end of the study period.

Author Response

Dear Reviewer,

I am sorry that the previous review was unsatisfactory., and I thank you for the time spent on this manuscript.

The rationale of our study aim is the following:

CR is especially recommended in the elderly because of their higher risk factors for CVDs and because CR has been shown to be effective also in this population.

However, the participation of elderly patients in CR is still limited, and the lack of HCP referral plays an important role in this. Nonetheless, CR in the elderly population must be adapted to numerous variables, and treatments that are too burdensome in the face of negligible benefits must be avoided.

Therefore, it has been recently recommended to collect further evidence on the efficacy of CR in the elderly, as this could encourage clinician referral and increase participation.

Therefore, we wanted to study the cohort of our patients to see whether and how their characteristics have changed over time and to see whether and how responses to CR have changed.

I explained this rationale in more detail in the Introduction (lines 112-125) and three references were added (n.4, 5 and 7); I hope it is clearer now.

Furthermore, if you're interested in the efficacy of CR in elderly patients, why not look at this group in isolation? I fail to see the value in assessing whether the average patient was older at the end of the study period.

Dear Reviewer, the choice to examine the entire cohort of patients included in the CR at our hospital over a long time span was based on our wanting to see whether these patients had actually changed over the years. In fact, referral may have been limited to the younger individuals, but our data did not bear this out, and more and more elderly individuals have had access to CR. In our opinion, this information could be useful to the readers. Nevertheless, we investigated the results obtained on clinically relevant outcomes and verified that, although the population included in the CR program changed over time, the results remained satisfactory, thus contributing to the recommendation to produce data on the efficacy of CR in the elderly population.

Reviewer 2 Report

No one

Author Response

Thank you very much for your appreciation. We have the manuscript checked for spelling again. Thank you.